# Theoretical Investigation of the Coronavirus SARS-CoV-2 (COVID-19) Infection Mechanism and Selectivity

**DOI:** 10.3390/molecules27072080

**Published:** 2022-03-24

**Authors:** Iga Biskupek, Adam Sieradzan, Cezary Czaplewski, Adam Liwo, Adam Lesner, Artur Giełdoń

**Affiliations:** Faculty of Chemistry, University of Gdansk, ul. Wita Stwosza 63, 80-308 Gdansk, Poland; dominika.biskupek@phdstud.ug.edu.pl (I.B.); adam.sieradzan@ug.edu.pl (A.S.); cezary.czaplewski@ug.edu.pl (C.C.); adam.liwo@ug.edu.pl (A.L.); adam.lesner@ug.edu.pl (A.L.)

**Keywords:** SARS-CoV-2, COVID-19, ACE2, molecular modeling, UNRES

## Abstract

The SARS-CoV-2 virus, commonly known as COVID-19, first occurred in December 2019 in Wuhan, Hubei Province, China. Since then, it has become a tremendous threat to human health. With a pandemic threat, it is in the significant interest of the scientific world to establish its method of infection. In this manuscript, we combine knowledge of the infection mechanism with theoretical methods to answer the question of the virus’s selectivity. We proposed a two-stage infection mechanism. In the first step, the virus interacts with the ACE2 receptor, with the “proper strength”. When the interaction is too strong, the virus will remain in an “improper position”; if the interaction is too weak, the virus will “run away” from the cell. We also indicated three residues (positions 30, 31, and 353) located on the ACE2 protein-binding interface, which seems to be crucial for successful infection. Our results indicate that these residues are necessary for the initiation of the infection process.

## 1. Introduction

The SARS-CoV-2 virus first occurred in December 2019 in Wuhan, Hubei Province, China. The pathogen was identified in January 2020 and is commonly known as COVID-19 [1]. The virus can be very dangerous since it can cause damage to the lungs, heart, kidney, liver, and skin, as well as the central nervous system [2,3,4]. Since the virus can be very dangerous, some safety rules have had to be applied to minimize the infection’s impact [5,6]. Multiple sequence alignment of the CoV genome indicates that these viruses share a high sequence similarity of about 60% among the nonstructural proteins and about 45% among the structural proteins. Additionally, human and bat SARS-CoV-2 viruses share about a 96% similarity [7]. The infection mechanism of the virus is via the angiotensin-converting enzyme 2 (ACE2) [8,9,10,11,12]. The ACE2 genome was identified in 2000 [13,14] and is a crucial component of the renin-angiotensin system (RAS) [15,16], mediating numerous systemic and local effects in the cardiovascular system [14,17] and protecting against a variety of pulmonary diseases [14,18]. The ACE2 genome shares a high sequence homology in the animal kingdom [19]. The high expression of this protein in both the cardiovascular and respiratory systems makes them vulnerable to infection. The other known infection mechanisms of coronaviruses are via the dipeptidyl peptidase 4 (DP4)-MERS-CoV and aminopeptidase N (APN)-HCoV-229E [20,21]. The second virus, identified as a disease in 2003, was SARS-CoV [22]. This virus also interacts with the ACE2 receptor [20]. It is known that the catalytic activity of the ACE2, APN, and DPP4 peptidases is not required for virus entry, and the co-expression of other host proteases allows efficient viral entry. Additionally, evolutionary conservation of these receptors may permit interspecies transmissions [23].

There are many substances that can increase ACE2 expression in the cells. One of the substances most commonly used is ibuprofen [24,25,26]. From the medical point of view, that information is crucial since it will decrease the prognosis for successful treatment. On the other hand, some substances were reported as an inhibitor of different points on the SARS-CoV-2 pathway [27,28,29,30]. The other possible treatment pathways are via inhibition of the RNA-dependent RNA polymerase, inhibition of protease enzyme, the regulation of cytokine formation, inhibition of CD147, among others [31,32]. Wang and co-workers published a structure of the coronavirus spike receptor-binding domain (which is a glycoprotein [33]), complexed with its receptor, ACE2 (PDB code 6LZG) [34], which revealed the infection mechanism at the molecular level [35]. At the same time, Gou and co-workers published a work that indicated SARS-CoV-2 selectivity against certain species. They checked SARS-CoV-2 vulnerability for bats, pigs, civets, and mice [36]. It appeared that only the mouse is fully resistant to the infection. Damas et al. made a more detailed sequence analysis of the ACE2 receptor interface. According to their results, the human infection rate is very high. The same situation is seen with a variety of monkeys and apes. In the case of dogs (in terms of the dependency of the race), it is either low or very low. The same situation is seen with the bat and mouse models. The authors also reported that bats have the highest diversity of beta coronaviruses among mammals [37]. Additionally, the WHO declared that there is no evidence that SARS-CoV-2 can infect pets; however, some exceptions were reported by the American Veterinary Medical Association [38].

In this work, we used molecular modeling techniques to identify the crucial residues responsible for the interaction of the SARS-CoV-2 spike receptor and the ACE2 protein. The detailed analysis of the indicated interface may give some information on the potential successful inhibitor of the virus. To verify the above statement, we performed a molecular dynamics simulation, as implemented in the UNRES force field [39].

## 2. Methods

The coronavirus spike receptor-binding domain, complexed with its receptor ACE2 (PDB code 6LZG) [34], was used as a template for modeling. In the next step, the sequence of ACE2 receptors of the indicated species was obtained from the Swiss Prot Data Base [40]. In this work, we used the following sequences with Swiss Prot codes: human-Q9BYF1 [41], bat-G1PXH7, pig-K7GLM4, civet-Q56NL1, mouse-Q3URC9 [42], and dog-F1P7C5.

All sequences were aligned using the Clustal software [43]. In the next step, the residues located on the ACE2-spike receptor-binding domain were identified using the RasMol AB software [44] (See Figure 1). To obtain the models of the ACE2 receptor of selected species, the point mutations were made using the UCSF Chimera software [45]. The most probable or the most fitting rotamer was applied during the computer mutation with the rotamer library, as implemented in the UCSF Chimera software [46]. The electrostatic surface was calculated using APBS, the adaptive Poisson-Boltzmann solver [47], as implemented in the PyMol (Delano Scientific, San Carlos, CA, USA) plugin. The continuum electrostatics calculations used for the surface as drawn were calculated using the PDB2PQR software [48].

The newly built SARS-CoV-2 spike receptor-ACE2 (human, bat, and mouse) models were simulated in the UNRES forcefield. UNRES is a united-residue forcefield designed to simulate peptides and proteins. The protein chain is represented by a sequence of C_α_ atoms, connected via virtual bonds and the united peptide groups located in the middle, with attached united sidechains. Twenty-four independent molecular dynamics runs were carried out for the three selected complexes. Eight million MD steps of 1 MTU (molecular time unit) were computed. The MTU used in UNRES MD amounts to 48.9 fs, which leads to 391 ns for the simulation time of each trajectory. However, it should be noted that because of averaging over the secondary degrees of freedom, the time scale of UNRES MD is extended by 1000–10,000 times compared to the all-atom time scale. The discrepancy between the simulation and the biological time scale restricts the applicability of this technique for solving concrete biological problems [49,50,51].

## 3. Results

To identify the most promising interaction site for the molecule that will inhibit the angiotensin-converting enzyme 2 (ACE2) and SARS-CoV-2 spike receptor domain, we performed a detailed analysis of the indicated binding interface. Four fragments of ACE2 are located on the spike receptor interface. These fragments consist of residues in the range of 19–49 (I), 79–83 (II), 324–330 (III), and 352–356 (IV) (see Figure 1 and Figure 2). From the spike receptor site, there are also four areas of interaction: 417, 455–456, 475–478, and 486–505. In the ACE2 protein, the largest area of interactions consists of the first α-helix, the C-terminal part of the second α-helix, one β –turn, and one loop. From the SARS-CoV-2 spike receptor, there are two twisted β-sheets and two loops. To distinguish the differences between species and SARS-CoV-2 infection capability, we performed a multiple sequence alignment of the ACE2 (see Figure 1). In general, ACE2 is highly genetically conserved among all analyzed species. Therefore, we decided to distinguish two types of interactions: common and not crucial for SARS-CoV-2 selectivity, and unique and potentially crucial for virus selectivity. For clarity, to distinguish the differences between ACE2 and SARS-CoV-2 spike receptor residues, the following abbreviations will be used in order, as ACE2 and SR, respectively. Crucial interactions are pointed out in the manuscript (see Figure 2). For a 2D plot see Appendix A).

## 4. Common Interactions

Firstly, S19_(ACE2),_ located on the N-terminal part of the first α-helix, can form a hydrogen bond with the carboxyl group from the peptide bond of A475_(SR)_ (also observed by Wang et al. [52]).

Secondly, Q24_(ACE2)_ interacts with N487_(SR)_ (also observed by Wang et al. [52] and Zhai et al. [53]). Three types of amino acid residues are located in this position (see Figure 1). In humans and mice, there is an amide; in civets, dogs, and pigs, there is hydrophobic residue (L); in bats, there is a base residue (K). Since humans and mice have the same type of residue in the ACE2 protein, and we know that mice cannot be infected, we may suppose that this interaction is not crucial for infection. The methyl group from T27_(ACE2)_ is located in between three hydrophobic residues, F456_(SR)_, Y473_(SR),_ and Y489_(SR)_ (as observed also by Zhai et al. [53]). The hydroxyl group is directed toward the solvent. Threonine is present in all the analyzed species, except the bat, which has isoleucine. The indicated area of the SARS-CoV-2 spike receptor will favor the hydrophobic residues, and the interaction with only the methyl group will be much weaker. Therefore, we may speculate that the virus’s origin was from a bat. H34_(ACE2)_ interacts with Y453_(SR)_ and L455_(SR)_ (also observed by Zhai et al. [53]). Although we have a variety of residues located at this point in the sequence, we do not suppose that this residue is crucial for the SARS-CoV-2 infection capability. In humans, civets, and dogs, there is an aromatic residue; since dogs cannot be infected, it seems that this is not a crucial infection factor. In bats, we have serine, which is a small residue, and in pigs, we have leucine (see Figure 1). Both species can be infected. E35_(ACE2)_, D38_(ACE2),_ and Q42_(ACE2)_ can form a hydrogen bond with Q493_(SR)_, Y449_(SR),_ and Q498_(SR)_, respectively (as also observed by Wang et al. [52] and Zhai et al. [53] and as identified by Damas at al. [37]). These interactions are present in almost all analyzed species; therefore, we may speculate that they are not crucial for the SARS-CoV-2 infection capability. The aromatic residue in position 41 (Y and H) is genetically well-preserved among all analyzed species (see Figure 1) and it is directed toward Q498_(SR)_, T500_(SR),_ and N501_(SR)_ (see Figure 2) (also observed by Zhai et al. [53]). This region was also identified by Day et al. [54] as an interaction region between the SARS-CoV-2 spike receptor and ACE2 protein. Since this interaction is not specific among all analyzed species, we may speculate that it is not crucial for the infection mechanism. Three residues from the C-terminal part of the second α-helix, namely L79_(ACE2)_, M82_(ACE2)_ and Y83_(SR)_ create an interaction cluster with N487_(SR)_, F486_(SR)_, and Y489_(SR)_ (also observed by Zhai et al. [53]). This region was also identified by Nelson et al. [55]. In this case, there are two sequential differences between humans and mice, in position 79 (L/T) and in position 83 (Y/F). On the other hand, a tyrosine residue is present. If we compare the interaction energy of the indicated residues [56], it appears that the difference is only about 2kcal/mol in favor of the mouse. This is definitely too low a level to be crucial for stabilizing the interaction between ACE2 and the SARS-CoV-2 spike receptor. The first look at the 324–330 fragment of the ACE2 receptor (see Figure 1) may suggest that position 329 (E/A)_(ACE2)_ may play a crucial role in the interaction of the investigated proteins. Those residues are located in the small helix, directed toward the spike receptor. The conformation of the backbone suggests that only those residues in positions 326_(ACE2)_ and 330_(ACE2)_ can form a strong interaction. At this point, there is no difference between humans and mice. In position 326_(ACE2),_ the glycine residue is present and N330_(ACE2)_ can form a weak interaction with T500_(SR)_. On the other hand, even if we could consider the interactions via a water molecule, there is no partner for a salt bridge for E329_(ACE2)_. Therefore, we may speculate that this mutation is also not important in the interaction between ACE2 and the SARS-CoV-2 spike receptor.

## 5. Unique Interactions

One of the crucial interactions that stabilize the protein-protein interface is D30_(ACE2)_ and K417_(SR)_ (see Figure 2). This salt bridge can be created in almost all analyzed species (except the mouse). Aspartic acid is present only in the human protein. In the other species, glutamic acid is present, which can also create the indicated salt bridge. In the mouse ACE2 protein at this position, asparagine is present, which is not capable of creating the salt bridge. The second important residue, located in position 31_(ACE2)_ (see Figure 1) on the ACE2 interface, is (K/N/T) (human, dog, pig/mouse, bat/civet). The 6LZG PDB structure revealed that in this position, the most preferable residue is lysine since it can create a salt bridge with E484_(SR)_. The most surprising fact is that mice (which cannot be infected) and bats (which can be infected and are considered to be a source/reservoir of the virus [57]) demonstrate an asparagine residue. This fact may be a counter-proof for the theory that bats are the source of the infection. Another interesting fact is that dogs show the same residues as the other species that can be infected. We may speculate that this is the reason why some infections in dogs were observed by the American Veterinary Medical Association [38]. K353_(ACE2)_ is located in the hydrophobic nest, consisting of Y495_(SR)_, F497_(SR),_ and Y505_(SR)_ (see Figure 1). At first glance, the histidine residue, which is present only in the mouse protein (see Figure 1), should be the most preferred one, since it can create π-π interactions. In this case, it is not true since large residues with limited conformational flexibility, like phenylalanine, tyrosine, tryptophan, and histidine, will, in fact, push the spike receptor away instead of interacting with it.

## 6. Electrostatic Surface Analysis

To verify the SARS-CoV-2 spike protein’s fit to the ACE2 protein, we performed an analysis of the electrostatic surface for the investigated proteins. The surface was calculated using the Poisson–Boltzmann continuum solvation models, at pH = 7, as implemented in the PDB2PQR software [48] and visualized in PyMol (Delano Scientific, San Carlos, CA, USA) (see Figure 3). It appeared that the best electrostatic fit could be achieved for the virus receptor to the human protein. K417_(SR),_ of the virus spike protein, can adapt well to a small cavity with D30_(ACE2)_ in the center. Simultaneously, K31_(ACE2)_ can be located in the cavity on the spike protein’s surface with E484_(SR)_ (see Figure 3). The relevant area was marked as a horizontal ellipse (see Figure 3) and has been described in Section 5: Unique Interactions. At first glance, it can be seen that in the human protein, the positive and negative charges are perfectly matched in the horizontal ellipsoid. In the case of the bat protein, we have one that attracts and one that repels interaction. Two repelling interactions that are present in the mouse protein protect the mouse from infection. The other two areas of interaction, marked on the visualization of electrostatic potential, seem not to be important for the interaction between ACE2 and the SARS-CoV-2 spike protein.

## 7. Molecular Dynamics Simulation

To investigate the stability of the ACE2-spike receptor interface, we performed a molecular dynamics simulation, as implemented in the UNRES forcefield. To distinguish differences between the species, we constructed models of the ACE2 protein in humans, mice, and bats and simulated the dynamics with the SARS-CoV-2 spike receptor domain, keeping the starting conformation as present in the 6LZG PDB structure. As a result of the simulation, we observed three stable conformations in the simulated systems. The first conformation was similar to the initial position of the spike receptor against the ACE2 protein. In this case, the spike receptor loop (473–489) was located near the N-terminal part of the first α-helix of ACE2 (see Figure 2). After the simulation, this conformation was only present in the bat ACE2 model. In the second conformation (not shown), the spike receptor loop (473–489) was located near the C-terminal part of the first α-helix. Since this part is located in the cellular membrane, this meant that the MD trajectory failed. In the third conformation, the (473–489) loop was located near the third α-helix and it was directed toward the interior of the protein (see Figure 2). Since this part of the ACE2 protein is situated in the “front of the cell”, we speculated that this conformation is responsible for virus recognition. Some trajectories finished in an unspecified conformation. For a detailed statistical analysis, see Table 1.

More than half of the simulations (see Table 1) of the human system ended with a spike receptor located on the top of the ACE2 protein (see Figure 2). This state was also observed in bats and mice; however, in this case, the percentage was much lower. We could, thus, speculate that this conformation is responsible for the first step on the infection pathway. Therefore, we decided to analyze it in detail.

## 8. Interactions at the First Stage of the Infection

L455_(SR)_ was creating weak hydrophobic interactions with F28_(ACE2)_ and Y83_(ACE2)_ and also with Q24_(ACE2)_. Phenylalanine and tyrosine are well conserved among all analyzed species (see Figure 1). Conversely, the residue in position 24 is unique. In the civet, dog, and pig, there is a leucine residue that should be the most optimal solution for the creation of hydrophobic interaction; however, the energy difference, in this case, is about 1 kcal/mol [56]. Since there is no marked difference in the interaction energy, we may suspect that this interaction is not crucial to the infection mechanism. A hydrophobic cluster was created by Y495_(SR)_ and Y505_(SR),_ and F28_(ACE2)_ and L79_(ACE2)_ (see Figure 2). Phenylalanine and leucine are highly conserved among almost all the examined species (L→T mutation is present in the mouse) (see Figure 1). Those residues create a good fitting interface. Additionally, a cation-π interaction can be found between R403_(SR)_ and F28_(ACE2)_. We found that Y449_(SR)_ created an interaction with F72_(ACE2)_ and E75_(ACE2)_. The sequence analysis showed that both residues are highly conserved among almost all analyzed species (see Figure 1) (except in the bat, in which E→Q is present). Since we know that the mouse is SARS-CoV-2 resistant, we did not consider that interaction to be crucial to the infection mechanism. E484_(SR)_ can create a hydrogen bond interaction with N103_(ACE2)_ and Q81_(ACE2)_ (see Figure 2). This interaction is preserved in dogs and pigs, in which the lysine residue is present. The creation of the salt bridge can reinforce the interaction during the first step of the infection mechanism and prevail to carry this further. Another weak and common interaction can be observed between V483_(SR)_ and A193_(ACE2)_ and V107_(ACE2)_. Position 107 is highly conserved since the only difference can be observed in the case of the mouse (alanine) and pig (glycine) models (see Figure 1). A common interaction between F486_(SR)_ and three glutamine residues (98, 101, 102)_(ACE2)_ was found in the identified stable state. These glutamine residues are highly conserved among almost all examined species (except in the dog) in which histidine and arginine residues are present. They created a nest in which F486 is placed. At first glance, the interaction of N487_(SR)_ with Q86_(ACE2)_ is not important since Q is present in humans, bats, and mice. We know that mice cannot be infected by SARS-CoV-2. However, in the civet, we have alanine; in dogs and pigs, we have acidic residues. Acidic residues will reinforce the interaction by about 4 kcal/mol in comparison to an amide–amide interaction, so we may expect that dogs and pigs should be more vulnerable to infection (although they are not). M82_(ACE2)_ interacts with a backbone of L492_(SR)_ and Q493_(SR)_ in the spike receptor of the virus. In the other species, we have threonine and serine residues. Therefore, we could expect that all species (except humans) are vulnerable to infection. A large hydrophobic cluster created by F456_(SR)_, Y489_(SR)_, M82_(ACE2)_, P84_(ACE2),_ and L85_(ACE2)_ can be identified (see Figure 2). It seems that this hydrophobic cluster plays some role in the first step of the infection mechanism. The only difference among all the analyzed species is that in position 84, the mouse model has serine residue. Since serine will favor interactions with water, this may impede the attachment process of the spike protein to the ACE2 receptor. In the vicinity of this interface, we have two proline residues (82 and 84), which are not directly involved in the SR–ACE2 interface. We may speculate that the presence of that proline is not a coincidence but is rather an evolutionary adjustment. A hydrogen bond interaction was created by the S494_(SR)_ sidechain with an oxygen atom from the peptide bond between E75_(ACE2)_ and Q76_(ACE2)_. Those residues are highly conserved among all analyzed species; however, in this case, the interaction is via the peptide bond and not the sidechain. Q498_(SR)_ is interacting with E35_(ACE2)_, K68_(ACE2),_ and F72_(ACE2)_ (see Figure 2). The residues in positions 68 and 72 are highly conserved among all analyzed species (see Figure 1). E35 is also well conserved; however, in the bat model, E→K mutation can be found. As a consequence, this interaction is reinforced by about 9 kcal/mol [56]. We may speculate that this mutation allows the bat to play the role of a virus carrier [58]. K417_(SR)_ is located between T20_(ACE2)_ and Q24_(ACE2)_. The residue in position 24 is not well conserved among all the analyzed species. We found the following mutations: L-(civet, dog and pig), Q-human, K-bat, and N-mouse. It seems that from the energetic point of view, the most preferred interaction is in humans and mice. However, conversely to humans, this is the only strong interaction in the mouse model.

## 9. Conclusions

In this paper, we described the results of a molecular dynamics simulation of the SARS-CoV-2 spike receptor and ACE2 protein of selected species (mouse, bat, and human). We used a coronavirus (SARS-CoV-2) spike receptor-binding domain complexed with its receptor ACE2 (PDB code, 6LZG) [34] as a template for modeling. The detailed analysis of the interface confirmed the published data that the SARS-CoV-2 spike receptor and ACE2 fit well with each other. The most important interactions that seem to play a key role in the stabilization of the protein-protein interface, which is crucial in the infection mechanism, are two salt bridges, namely, D30_(ACE2)_-K417_(SR)_ and K31_(ACE2)_-E484_(SR)_. With these interactions, the virus can create a stable “connection” to the infected cell. All-atom simulations confirmed this theory. The molecular dynamics simulations performed by Ali et al. [59] and Piplani et al. [60] showed very limited fluctuations in the region around K417. Additionally, E484 plays an important role in the orientation of the spike receptor toward ACE2. The results of those simulations confirmed the stability of the ACE2 spike receptor interface. Due to the high affinity of the SARS-CoV-2 spike receptor and ACE2 interface, using all-atom simulation, it is possible to perform only a local search and indicate all dynamic details. To explore this more widely, a very long simulation should be performed or a coarse-grained forcefield should be used.

With our results, we were able to identify a stable and temporary interface located on the “front” of the spike receptor. We speculate that this conformation is the first step of the infection cascade. However, in terms of the dependency on the species, this appears to be different. The first and, it seems, the most important condition is that the interaction has to have an “ideal” strength. According to our results, the mouse model is not being infected since the interaction on the first step is not strong enough to hold the virus near to the cell membrane. On the contrary, our analysis of the ACE2 sequence of the dog allows us to conclude that, in this case, the interaction is very strong, and the virus remains attached to ACE2 and does not travel further. We may speculate that the virus is being attached to the ACE2 receptor and does not cause infection, or that dog infection is extremely rare. A piece of partial evidence for this hypothesis can be found in the study published by Sit and co-workers [61]. They found that the human-to-animal transmission of SARS-CoV-2 is possible; however, it is still unknown if dogs can transmit the virus to other animals or back to humans. Nonetheless, the authors detected antibody responses using plague-reduction neutralization assays. One of the possible explanations for this finding is that the dog organism had enough time to create antibodies since the virus was attached to the “top part” of the ACE2 receptor and remained in the body for long enough. In our results, we found two pieces of structural evidence for this hypothesis. The first piece of evidence is the interaction of N487_(SR)_ with Q86_(ACE2)_. In dogs and pigs, these are acidic residues and they will reinforce the interaction by about 4 kcal/mol, in comparison to amide–amide interaction. The second piece of evidence is in the interaction of M82_(ACE2)_ and L492_(SR)_ and Q493_(SR)_. With this interaction, we could expect that the only species that should be SARS-CoV-2 resistant is humans (but this is not the case).

## Figures and Tables

**Figure 1 molecules-27-02080-f001:**
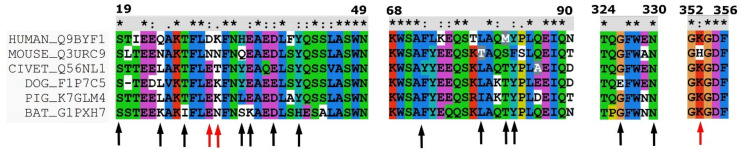
Multiple sequence alignment of the angiotensin-converting enzyme 2 (ACE2) in human-Q9BYF1, mouse-Q3URC9, civet-Q56NL1, dog-F1P7C5, pig-K7GLM4, and bat-G1PXH7. Common and unique interaction places are indicated as black and red arrows, respectively. Stars and dots indicate the sequence similarity, as presented in the Clustal software.

**Figure 2 molecules-27-02080-f002:**
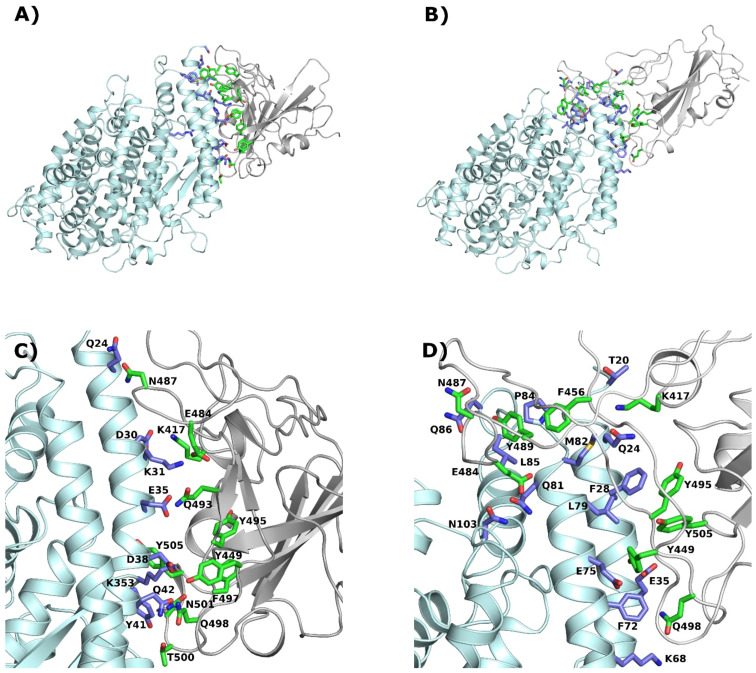
Structural details of the interface between the spike receptor of SARS-CoV-2 and the Human ACE2 receptor. ACE2 is colored pale cyan, while SR is colored gray. The interacting residues were colored blue and green, respectively. (**A**,**C**) 6LZG crystal structure; (**B**,**D**), the results of MD simulation.

**Figure 3 molecules-27-02080-f003:**
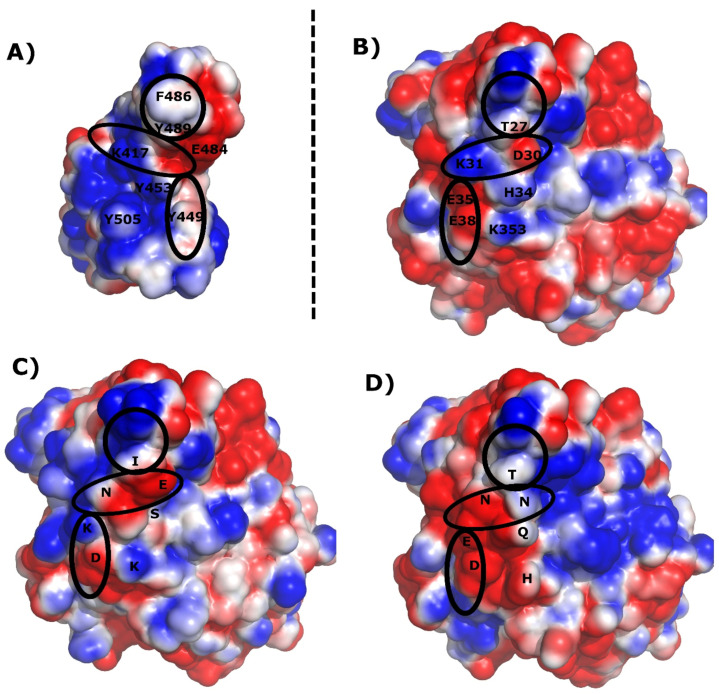
Visualization of the electrostatic potential of the SARS CoV-2 spike receptor, and the different variants of the ACE2 receptor. The blue region shows the location of positive electrostatic potential, while the red region is the location of negative electrostatic potential. Calculations were performed at pH = 7.0. (**A**) SARS CoV-2 spike receptor, (**B**) human ACE2, (**C**) bat ACE2, (**D**) mouse ACE2.

**Table 1 molecules-27-02080-t001:** The statistical analysis of the conformation, described in the text as the third one, achieved during the molecular dynamics simulation.

Species/Description	Human	Bat	Mouse
Number of simulations	16/24	1/24	2/24
Percentage success	67%	4%	8%

## Data Availability

Not applicable.

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
