# Peer review of "Theoretical Investigation of the Coronavirus SARS-CoV-2 (COVID-19) Infection Mechanism and Selectivity"

_molecules, 2022, doi:10.3390/molecules27072080_

Round 1
Reviewer 1 Report
Dear editor,
Have a nice day. The authors reported molecular dynamics simulation of the SARS‐CoV‐2 spike receptor and ACE2 protein of some species (mouse, bat, and hu-269 man). coronavirus spike receptor-binding domain complexed with its receptor 270 ACE2 (PDB code 6LZG) was used as a template for modeling. Three residues (positions 30, 31, and 353) located on the ACE2 protein binding interface were expected to be crucial for the successful infection.
The manuscript needs major modification prior its publication as follows.
- Protein-protein docking should be carried out to confirm the results of MD simulation.
- Molecular Mechanics Poisson–Boltzmann Surface Area (MM-PBSA) should be carried out to calculate the free binding energy of the SARS‐CoV‐2 spike- ACE2 protein complex. The MM-PBSA is an advanced method that can examine the binding between a protein and a specific ligand through computing the exact binding free energy of the ligand-protein complex over a determined time.
- RMSD and RMSF should be carried for SARS‐CoV‐2 spike- ACE2 protein complex.
- 2D interactions should be presented to clarify the crucial amino acids incorporated in the binding process.
Best regards
Author Response
Dear Reviewer,
The idea for this paper, comes spontaneously. We just wanted to see, what will happen with spike - ACE2 complex during the simulation. It appeared, that more than half of the systems, ended at a specific point. The question was why? Then we found a nice paper on animal infection study. From a statistical point of view, the results were completely different. This is, why we proposed a two-stage recognition mechanism. Anyway, when the virus comes near the cell it cannot be bounced away…
The calculation of MM-PBSA is a fantastic idea. However, if we look at the running section in AMBER manual:
http://ambermd.org/tutorials/advanced/tutorial3/section2.htm
We will need 400 CPUs for every simulation to do it right. Three or five systems in two conformations. It means, that we will need 4000 CPUs to calculate everything. With the analysis, it is a work for two months. The idea is fantastic and for sure the amount of work to be done, justify it for a separate paper.
The idea of coarse grain simulation is to have a quick search of the hypersurface of the potential energy. The disadvantage is, that we are losing the resolution. In the case of UNRES it is more than 4A. As you pointed, the fluctuations are high, but this is the way how UNRES works. On the contrary to the all-atom force field with which you can perform a local search like
(DOI:10.1371/journal.pone.0161526 or DOI: 10.1002/cbic.200800324)
The idea of 2d plot is fantastic and we included it in the supplementary data using logplot software.
Additionally, all options from the input file were included in supplementary data.
Sincerely
Artur Gieldon
Reviewer 2 Report
This manuscript describes the results of a homology modeling study to explain the species specificity in the infection of SARS-CoV-2.
Unfortunately, this paper does not reach the standard level of scientific papers because the results and conclusions were not derived from clear evidence.
Some specific points:
(1) The species specificity has already been discussed thoroughly, both experimentally and theoretically. As theoretical papers on this topics, the following two are found in the recent list, and they are not cited in this paper.
Damas J, et al. Broad host range of SARS-CoV-2 predicted by comparative and structural analysis of ACE2 in vertebrates. Proc Natl Acad Sci U S A. 117, 22311 (2020).
Piplani, S. et al. In silico comparison of SARS-CoV-2 spike protein-ACE2 binding affinities across species and implications for virus origin. Sci Rep 11, 13063 (2021).
(2) The molecular dynamics simulations using the UNRES force field resulted in the collapsed interface shown in Fig. 2B, D. The UNRES with neither explicit side-chains nor explicit solvent is not suitable for the simulation of protein-protein interactions.
Author Response
Dear Reviewer,
The idea for this paper, comes spontaneously. We just wanted to see, what will happen with spike - ACE2 complex during the simulation. It appeared, that more than half of the systems, ended at a specific point. The question was why? Then we found a nice paper on animal infection study. From a statistical point of view, the results were completely different. This is, why we proposed a two-stage recognition mechanism. Anyway, when the virus comes near the cell it cannot be bounced away…
Unfortunately, there is no guide, that this and that paper should be included in the manuscript about a specific subject. However, the papers, you pointed out are very nice, and now they are included in the manuscript.
The second remark is about the methodology. Unfortunately, I cannot agree with you.
1. The implicit solvent is included in UNRES energy function. The sidechain – sidechain (crucial in protein – protein interactions) free energy interactions were calibrated to include if they are surrounded by water or other residues. (J. Comput.Chem., 1997, 18, 849-873).
2. There are many coarse grain force fields, which are used for protein – protein interactions. The obtained results matched well with the experimental data.
CABS, doi:10.3390/ijms22147341
PACSAB doi:10.1021/acs.jctc.5b00660
3. UNRES force field is able to predict protein – protein interactions and as proof see the results from CASP experiment (doi:10.1002/prot.26222)
4. UNRES was already validated in a prediction of protein – protein interactions and it was able to predict the mutation effect on the stability of the protein complex. (Proteins: Struct. Funct. Bioinfo., 2015, 83, 1414–1426).
The available MD simulations show only a local search of the spike - ACE2 interface. With coarse grain simulation we were able to perform a more wide search.
sincerely,
Artur Gieldon
Reviewer 3 Report
It is a good idea to combine the experimental fact and the computation to elucidate the secret of the infection of Covid 19-SARS2.
It is recommended to publish this article with some minor revision.
First, it is better to describe the method detail for computation.
Second, the contrast detail of the docking analysis might be given to supplement the Figure 2 and 3, for these figures are not enough to give complete view of the interaction details.
Finally, please check the F489 for sure, if it should be Y489 instead?
Author Response
Dear Reviewer,
The idea for this paper comes spontaneously. We just wanted to see, what will happen with spike - ACE2 complex during the simulation. It appeared, that more than half of the systems, ended at a specific point. The question was why? Then we found a nice paper on animal infection study. From a statistical point of view, the results were completely different. This is, why we proposed a two-stage recognition mechanism. Anyway, when the virus comes near the cell it cannot be bounced away…
I realize that English is my second language and I know that it is far from perfection. Therefore, to upgrade it a little bit, I’m using grammarly extension to the word program. So I know that all comas, declensions and so on are correct. In most of the cases we had in our group a native-speaking prison, everything changed with COVID stuff….
The combination of an experiment and theory is the best possible solution for the research process. However, theoretical research is much cheaper. Sometimes it is also not possible to perform the experiment. The present force fields can mimic quite well a protein behavior, which was proven in thousands of manuscripts. Therefore, this paper should be treated as a hypothesis, proven by the theoretical experiment.
For a detailed and clear analysis, we decided to prepare a 2d figure and placed it in the supplementary data.
One of the most important thing in the scientific world is the ability to repeat an experiment done by someone else. Unfortunately, in the coarse grain force field, there are not so many things to report. So, to fulfill your request, all parameters from the input file were placed in the supplementary data.
Residue 489, a tragic error, thank you for pointing it.
Sincerely
Artur Gieldon

Reviewer 4 Report
The paper "Theoretical investigation of the coronavirus SARS‐CoV‐2 (COVID-19) infection mechanism and selectivity" is devoted to very significant topic, a mechanism of coronavirus spike protein binding to the ACE2 receptor. Despite paper is of clear interest; I feel some imrpovements are necessary.
- I feel that literature search could be performed more thoroughly. For example, one of the findings of the manuscript, the salt bridge between ASP30(ACE2) and LYS417 (SP) had been reported already in the paper (https://doi.org/10.1038/s41598-020-71188-3), which is not referenced in the manuscript. It is the first search result in Google using the terms "spike protein ACE2 receptor".
- The overall concept of the study needs an improvement. I feel it is insufficient just to describe the differences in separate amino acid residues and indicate whether the species are infected or not. Something more than that should be done; for example, I can suggest deriving some sort of model using discrete values corresponding to the presence or absence of some specific amino acids. The result predicted by the model is the infection outcome. It can be just simple logic model with zeros and unities.
- I also find hard to believe that the binding between spike protein and receptor depends only on three or even ten interactions between amino acid residues. Somehow the changing morphology should be taken into account. Moreover, as far as I know the spatial structure of spike protein changes when the binding takes place.
Author Response
Dear Reviewer,
The idea for this paper, comes spontaneously. We just wanted to see, what will happen with spike - ACE2 complex during the simulation. It appeared, that more than half of the systems, ended at a specific point. The question was why? Then we found a nice paper on animal infection study. From a statistical point of view, the results were completely different. This is, why we proposed a two-stage recognition mechanism. Anyway, when the virus comes near the cell it cannot be bounced away…
Unfortunately, there is no guide, that this and that paper should be included in the manuscript about a specific subject. However, the paper, you pointed out is very nice, and now it is included in the manuscript.
2. Well, I don’t have experience in the creation of the discrete model, but there is a big field in the cooperation, which we can start….
3. Of course it is not. And it was not the point of the paper. First, we analyzed in detail the ACE2 and spike interface and we confirm the literature data on the interface. There is an entire network of interactions between spike and ACE2. Some of the analyzed species can be infected, some cannot. Why, since they got a high sequence homology? As a conclusion, we pointed, out a few residues which seem to be crucial to maintain a stable interface. We have done the same, for the identified conformation. The results were not so obvious, why? The best idea (which I hope will be confirmed experimentally) is for a two-stage infection. If we have this the virus “would like” to shift the equilibrium to an infection-like state (pdb structure), simultaneously not being pushed away from the cell (structure found by us).
Sincerely
Artur Gieldon
Round 2
Reviewer 4 Report
The paper was improved since I've seen it for the first time, and Authors provided satisfactory answer to my concern. The topic chosen by Authors is indeed vast and complex and it would be just unfair to demand the exhaustive and comprehensive study of coronavirus binding to ACE2 receptor. I appreciate the originality and spontaneousness of the idea and look forward for new papers of Authors in this field.
Creating model is very simple: the "calculated outcome" is presented as a combination of the discrete parameters (which correspond to the presence or absence of the attribute, i.e. unities and zeroes). The combinations can be logical summation or multiplication. As a result, something like that can yield:
outcome = p1 | p2 | p3 | ... | pn
where symbol | stands for the logical operation. So, the problem is how to arrange the logical operators.
To be honest, I am not sure that it gonna give any useful results. I just don't like critisizing without supposing something. Perhaps (I'm even sure), there are much better ways to continue the research.
The idea of cooperation (if you mean the cooperation with me), alas, is as tempting as it is impossible. At least, not before the war ends.
Anyway, I do not see any good in further delay with the publication of this manuscript.